# Domain Invariant Q-Learning for model-free robust continuous control under visual distractions

## Abstract

End-to-end reinforcement learning on images showed significant performance progress in the recent years, especially with regularization to value estimation brought by data augmentation (Yarats et al., 2020). At the same time, domain randomization and representation learning helped push the limits of these algorithms in visually diverse environments, full of distractors and spurious noise, making RL more robust to unrelated visual features. We present DIQL, a method that combines risk invariant regularization and domain randomization to reduce out-of-distribution (OOD) generalization gap for temporal-difference learning. In this work, we draw a link by framing domain randomization as a richer extension of data augmentation to RL and support its generalized use. Our model-free approach improve baselines performances without the need of additional representation learning objectives and with limited additional computational cost. We show that DIQL outperforms existing methods on complex visuo-motor control environment with high visual perturbation. In particular, our approach achieves state-of the-art performance on the Distracting Control Suite benchmark, where we evaluate the robustness to a number of visual perturbators, as well as OOD generalization and extrapolation capabilities.

## 1 Introduction

Data augmentation is used extensively in computer vision models for regularization. One can not imagine reaching state-of-the-art performance on usual benchmarks without using a careful combination of transformation on images. Yet reinforcement learning lags behind on the usage of these techniques.

First, this stems from the high variance reinforcement learning suffers during training. This is especially true for off-policy algorithms such as Q-learning (Watkins & Dayan, 1992; Mnih et al., 2013), where noisy Q values caused by uncertainty induced an overestimation bias which renders training extremely difficult. A number of methods directly tackle the problem of overestimation with algorithmic or architectural changes to the value function (Van Hasselt et al., 2016; Wang et al., 2016; Bellemare et al., 2017; Kumar et al., 2021). Regularizing the value estimation with light data augmentation is another successful approach (Yarats et al., 2020; Laskin et al., 2020b) but extensive data augmentation in reinforcement means adding even more noise and can lead to difficult or unstable training (Hansen et al., 2021).

Secondly, standard computer vision tasks mostly focus on extracting high level semantic information from images or videos. Because classifying the content of an image is a substantially high level task, the class label is resilient to a lot of intense visual transformation of the image (e.g. geometric transformations, color distortion, kernel filtering, information deletion, mixing). Features such as exact position, relative organization and textures of entities in the image is usually not predictive of the class label and data augmentation pipelines take advantage of it. From a causal perspective, data augmentation performs *interventions* on the "style" variable which is not linked to the class label in the causal graph. It happens that for classification tasks the dimension of the style variable is much bigger than in visuo-motor control tasks where reinforcement learning is involved. Intuitively, we can change a lot of factors of variation in the visual aspect of a particular object without it

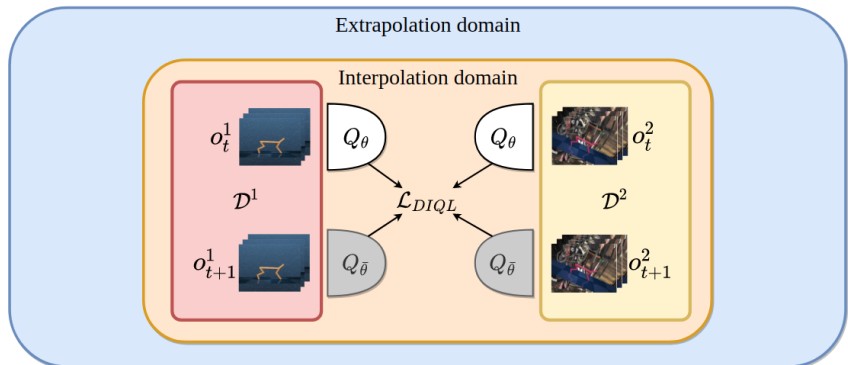

Figure 1: **Domain Invariant Q-Learning.** For training, DIQL uses 2 visually different domains (Interpolation domains) based on the same inner state of the environment to learn a Q-function that is invariant to spurious visual features. DIQL promotes risk extrapolation and prevents drastic collapse of the accuracy of the Q-function in out-of-distribution settings (Extrapolation domains).

being not recognizable anymore: we can still recognize a car on the street with very sparse and highly perturbed visual cues. With visuo-motor control, these factors of variation (dimensionality of the style variable) are in fewer number and less obvious. In particular, geometric deformations or occlusions could destroy crucial information for control such as relatives distances of objects in the image. Simple data augmentation, under the form of random shift, proved to be crucial for boosting RL performance (Yarats et al., 2020; 2021; Laskin et al., 2020b;a; Hansen et al., 2020) and is now used as a baseline in state-of-the-art methodologies. Though, it remains less clear which combination of image transformations is optimal for reinforcement learning (Hansen et al., 2021; Raileanu et al., 2021).

A related technique, **Domain randomization** (Tobin et al., 2017) was introduced in robotics to close the gap from simulation to real world, by randomizing dynamics and components in simulation. We argue in this paper that domain randomization is a more general case of data augmentation more suited for reinforcement learning that allows for finer control over visual factors of variation by directly changing the hidden state of the system in simulation. Contrary to data augmentation in general, domain randomization directly acts on the causal factors instead of adding uncorrelated noise to the observation, which could destroy useful information.

Starting from this observation, we present **Domain Invariant Q-Learning** (**DIQL**) for robust visuo-motor control under visual distractions. We show that DIQL is able to efficiently train an agent with visual generalization capabilities without losing on convergence speed and asymptotic performance on the original task. In particular, we derive a domain-invariant temporal-difference loss combining domain randomization and risk extrapolation (Figure 1). We show that domain randomization can be better integrated in reinforcement learning than is classically done in order to improve performance at very low cost. Our main contributions are:

- a novel methodology for robust visuomotor control based on temporal-difference learning on images using invariance principles,

- empirical results on the Distracting Control Suite benchmark (Stone et al., 2021) with state-of-the-art results on raw training performance and out-of-distribution (OOD) generalization under the hardest setting of dynamic distractions

## 2 PRELIMINARIES

**Value-based reinforcement learning**  We define the MDP $\mathcal{M} = \langle S, A, P, R, \gamma \rangle$ where $S$ is the set of states, $A$ the set of actions, $P$ the transition probability function, $R$ the reward function and $\gamma$ a discounting factor for future rewards. We also define the transition containing state, action, reward and next state at timestep $t$ as $\mathcal{T} = (S_t, A_t, r_t, S_{t+1})$. Reinforcement learning aims to maximize

the total reward received by the agent. Mathematically, it corresponds to finding a policy $\pi$ that maximizes the (discounted) expected return $\mathbb{E}_{\mathcal{T} \sim \pi}[\sum_{t=0}^{\infty} \gamma^t r(S_t, A_t, S_{t+1})]$

We are interested here in value-based reinforcement learning and in particular deep Q-learning, which approximates the state-action value function with a deep neural network to handle high-dimensional observations. The Q-network is optimized with gradient by minimizing the following temporal-difference loss:

$$J_Q(\theta) = \mathbb{E}_{(\mathbf{s}_t, \mathbf{a}_t, r_t, \mathbf{s}_{t+1}) \sim \mathcal{D}} \left[ \mathcal{L}_Q(\mathbf{s}_t, \mathbf{a}_t, r_t, \mathbf{s}_{t+1}) \right]$$
$$\text{where } \mathcal{L}_Q(\mathbf{s}_t, \mathbf{a}_t, r_t, \mathbf{s}_{t+1}) = (Q_\theta(\mathbf{s}_t, \mathbf{a}_t) - r_t + \gamma \max_{a'} Q_{\bar{\theta}}(\mathbf{s}_{t+1}, a'))^2 \tag{1}$$

$\bar{\theta}$ follow the trainable weights with an exponential moving average: $\bar{\theta} = \tau \theta + (1 - \tau)\bar{\theta}$ with $\tau$ the smoothing factor.

**Risk minimization and OOD generalization** The empirical risk associated to a model with parameters $\theta$ trained with a loss function $l$ with data $X$ situated on a domain $\mathcal{D}$ is the expected loss value of the model over that domain $R_l(\mathcal{D}; \theta) = \mathbb{E}_{X \sim \mathcal{D}}[l(X; \theta)]$. When working on out-of-distribution generalization and domain shift problems, we sometimes have multiple training domains. The true joint distribution $P(X, Y)$ of the data $X$ and labels/predictions $Y$ is often unknown and the simplest solution we can imagine to improve generalization is to minimize the averaged empirical risk above across data points and domains. Empirical risk minimization (ERM) is conservative and averages the training risk over the training distribution, without making further assumptions on out-of-distribution behavior. Formally, if the training dataset is composed of into multiple domains $\mathcal{D} = \{\mathcal{D}^i\}_{i \in 1, n}$, then using empirical risk minimization to optimize parameters $\theta$ of the model is equivalent to $\min_\theta \frac{1}{|\mathcal{D}|} \sum_i R(\mathcal{D}^i, \theta)$ where $R(\mathcal{D}^i, \theta)$ is the risk or cost function computed with data of domain $D^i$. However, ERM is naïve as the minimizer of the sum of the risks is not necessarily a minimizer of each subdomains. We can easily imagine a scenario where the joint objective of ERM is optimized even though the performance on one or more of the training domains is suboptimal. A fortiori, there is no guarantee of generalization to out-of-distribution domains at test time. Multiple improvements over ERM have been made, through causal inference (Peters et al., 2016) and invariant prediction (Arjovsky et al., 2019).

A more recent approach, V-REx (Krueger et al., 2021), directly tackles the issue of OOD generalization which is overlooked by previous methods. The problem is formulated as robust optimization and the objective is to minimize the risk $R^{OOD} = \max_{\mathcal{D}^i \in \mathcal{D}} R(\mathcal{D}^i))$. To practically minimize this worst-case risk, the empirical risk is complemented with a penalty based on the variance between training risks:

$$R_{V-REx}(\theta) = \sum_i R(\mathcal{D}^i, \theta) + \beta \text{Var}(\{R(\mathcal{D}^1, \theta), \dots, R(\mathcal{D}^n, \theta)\}) \tag{2}$$

This principle of invariance of risks promotes risk extrapolation and provably enables causal discovery.

## 3 METHOD

Our method builds upon deep Q-learning by regularizing and enriching the temporal-difference error with multi-domain and cross-domain risk minimization, taking fully advantage of domain randomization.

### 3.1 ON THE USE OF DOMAIN RANDOMIZATION

Data augmentation is a transformation applied a posteriori to an image. The transformation itself is most of the time completely independent from the image and agnostic to its content. Domain randomization on the other hand is acting directly inside an environment before an image is produced. Its subtlety lies in its ability to change appearances of scenes without changing its semantic content and inner state such as object classes and relative positions. Because our objective is to have an agent robust to visual perturbations, we will restrict the definition of domain randomization only to visual changes which do not causally change the inner state of the environment, usually called style variables or spurious features, as they are irrelevant for control.

Let's consider the causal graphical model between the environment and the Q-function with data augmentation or domain randomization in Figure 2. We are dealing with image observation, thus the true inner state $(\mathbf{s}_t, \bar{\mathbf{s}}_t)$ of the environment is hidden. $\mathbf{s}_t$ is the **controllable** part of the state whereas $\bar{\mathbf{s}}_t$ is control-irrelevant. We can think of $\bar{\mathbf{s}}_t$ as the part of the state containing the causes of spurious visual features in the observation $\mathbf{o}_t$ rendered by the environment. $Q_t$ denotes the true Q-value associated with $\mathbf{s}_t$ and $\mathbf{a}_t$ and is not affected by $\bar{\mathbf{s}}_t$. The difference between data augmentation and domain randomization is where the perturbation is injected. While data augmentation adds uncorrelated noise $X_{DA}$ directly to the observation, (visual) domain randomization $X_{DR}$ directly acts on $\bar{\mathbf{s}}_t$ independently from $\mathbf{s}_t$ **before rendering** $\mathbf{o}_t$. From a causal perspective, domain randomization performs do-interventions on the style variable $\bar{\mathbf{s}}_t$. This can help disentangling of the inner states $\mathbf{s}_t$ and $\bar{\mathbf{s}}_t$. In particular, this help the Q-function performing causal discovery by uncovering the effect of $\bar{\mathbf{s}}_t$ on $\mathbf{o}_t$. Then, by encouraging invariance to visual domain randomization, the Q-function is forced to ignore $\bar{\mathbf{s}}_t$ to make predictions and has to extract $\mathbf{s}_t$ and rely entirely on it. We call the resulting model a *domain-invariant* Q-function. This is not possible with data augmentation, as the noise is applied after rendering, which prevents disentagnling of the inner states. This motivates us to use domain randomization and to perform interventions with it at each step of learning.

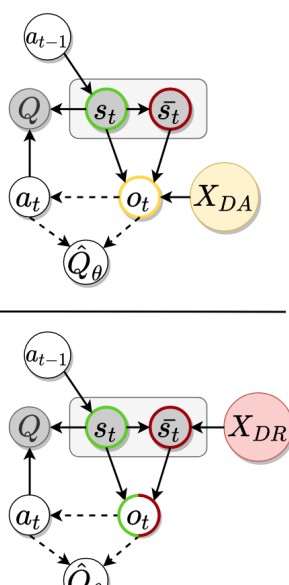

Figure 2: Causal Graphical Model of the environment and Q-function. (Top): data augmentation. (Bottom): domain randomization. Grey variables are hidden, white are visible. Dashed arrows indicate causality through inference by the policy and Q-value modules of RL, full arrows are invisible causal relationships.

## 3.2 DOMAIN INVARIANT Q-LEARNING

To maximize the utility of domain randomization and hopefully learn an invariant Q-function, we constantly generate 2 different views from the same scene at each time step in the environment (Figure 1). Formally, we can define the two views as two instances of the same scene where the style variable belongs to different domains, where $\mathbf{o}_t^1$ belongs to $\mathcal{D}^1$ and $\mathbf{o}_t^2$ to $\mathcal{D}^2$. Because accuracy of Q-values estimation is the main driver of performance for value-based model-free algorithms, our goal is to regularize the Q-function to make it invariant to visual perturbations irrelevant to reward and control.

We first regularize the Q-function by averaging the TD error over the two training domain previously defined, $\mathcal{D}^1$ and $\mathcal{D}^2$ by averaging the temporal-difference error over the two observations:

$$\mathcal{L}_{ERM} = \mathcal{L}_Q(\mathbf{o}_t^1, \mathbf{a}_t, r_t, \mathbf{o}_{t+1}^1) + \mathcal{L}_Q(\mathbf{o}_t^2, \mathbf{a}_t, r_t, \mathbf{o}_{t+1}^2) \tag{3}$$

where $\mathcal{L}_Q$ is defined in Equation 1. Action and rewards passed to the loss function are the same for both domains, as the unknown underlying state.

We add a domain translation term to the loss to further encourage domain invariance for the Q-function. For each additional TD error term, bootstrapped values are computed using the next observation $\mathbf{o}_{t+1}$ from the opposite domain of the current observation $\mathbf{o}_t$:

$$\mathcal{L}_{DT} = \mathcal{L}_Q(\mathbf{o}_t^1, \mathbf{a}_t, r_t, \mathbf{o}_{t+1}^2) + \mathcal{L}_Q(\mathbf{o}_t^2, \mathbf{a}_t, r_t, \mathbf{o}_{t+1}^1) \tag{4}$$

As mentioned in section 2, only averaging training risks over multiple domain is not enough to increase OOD generalization. In order to increase generalization to unseen domains, we regularize the losses with a variance minimization term that enforces invariance of training risks (here TD

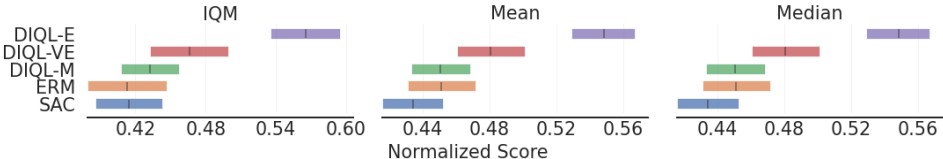

Figure 3: Evaluation metrics aggregated over all benchmarks, tasks, 21 episodes and 4 seeds at the end of training. Black stripes and colored bars represent aggregates and confidence interval computed with stratified bootstrap sampling using the rrliable library (Agarwal et al., 2021b).

losses). Motivated by Krueger et al. (2021), this term prevents one loss from dominating the others by "flattening" the risk landscape across domains.

$$\mathcal{L}_{Var} = Var\left(\left[L_Q(\mathbf{o}_t^{\mathbf{i}}, \mathbf{a}_t, r_t, \mathbf{o}_{t+1}^{\mathbf{j}})\right]_{(\mathbf{i},\mathbf{j})\in\{1,2\}}\right) \tag{5}$$

Putting all three losses together gives us our DIQL objective to train an invariant Q-function:

$$\mathcal{J}_Q^{DIQL}(\theta) = \mathbb{E}_{(\mathbf{o}_t^1, \mathbf{o}_{t+1}^1, \mathbf{o}_t^2, \mathbf{o}_{t+1}^2, \mathbf{a}_t, r_t)\sim\mathcal{B}}\left[\mathcal{L}_{ERM} + \mathcal{L}_{DT} + \beta\mathcal{L}_{Var}\right] \tag{6}$$

where $\beta$ controls the invariance of risks. $\beta$ at $\infty$ would force all risks to be equal but prevent minimization, while $\beta = 0$ recovers ERM applied to multiple domain to the Q-function.

Coming back to the causal inference framework, Krueger et al. (2021) show that under three specific hypothesis, equalizing training risks between domains is equivalent to performing causal discovery. The three hypothesis are the following: causes of the target variable are observed, training domains corresponds to interventions on $X$ and the Bayes error rate of the model is the same over all training points. The first hypothesis is verified: $Q_t$ has $\mathbf{s}_t$ and $\mathbf{a}_t$ as parents. Although $\mathbf{s}_t$ is hidden, all the information in $\mathbf{s}_t$ is contained in $\mathbf{o}_t$ due to the MDP framework. Thus, causes of $Q_t$ are observable. Second hypothesis is true by definition of domain randomization in our graphical model: domain randomization performs interventions on $\bar{\mathbf{s}}_t$. We perform random visual intervention, yet the controllable state of the environment is deterministic. In practice, we ensure that visual perturbations do not destroy useful information in the image: all of $\mathbf{s}_t$ is contained in $\mathbf{o}_t$ for all states. Because we are using a single Q-network with fixed capacity on the training domains, we deduce that the irreducible error, ie the Bayes error rate, is constant across all training samples. This validates the third hypothesis. Per Krueger et al. (2021) this means that equalizing training losses between each temporal-difference term in DIQL allows the Q-function to perform causal discovery. Thus, the Q-function can learn the hidden causes of the real Q-value and disentangle $\mathbf{s}_t$ from $\bar{\mathbf{s}}_t$: we have a *domain-invariant Q-function* by definition.

## 4 EXPERIMENTS

### 4.1 BENCHMARK AND BASELINES

We aim to evaluate the training stability, robustness and out-of-distribution generalization of our methodology with continuous control from images. We first present the Distracting Control Suite (DCS) that will serve to train our agent and benchmark for robustness to visual distractions. We detail our training procedures, baselines that will be used for comparison and evaluation protocol to quantify OOD generalization performance. Finally, we showcase aggregated and detailed evaluation measures on all six tasks of DCS, and present some ablations.

**Environment.** We use Distracting Control Suite (Stone et al., 2021) for our experiments. DCS is a variant of the Deepmind Control Suite where visual distractions are dynamically added to the rendered observations. The perturbations consists in the following non-exclusive dimension of variations: color randomization of physical bodies, background randomization with videos and camera movement. Distractions are dynamic, temporally consistent and continuous. Colors of bodies are

continuously changing at each time step. The background is displaying frame by frame a randomly selected video from the DAVIS dataset, which is played forward then backward to avoid discontinuities. The camera's orientation is rotating with a random angle at each step while keeping the agent in the field of view. As described in part 3, we produce two observations at each time step of the environment with two different visually randomized observations. In practice, instead of generating two randomized views, with generate a *clean* observation from the original environment and a *noisy* observation using a combination of camera, color and background distractions. This reduces computational burden and helps training by reducing variance in the data, which helps training as shown by Hansen et al. (2021). We define 5 variants of the environment, based on different combinations of distraction intensities and parameters. We use DCS with all three types of distractions applied together and varying the intensity value of the distractions. We refer to *Easy*, *Medium* and *Hard* benchmarks for 0.1, 0.2 and 0.3 intensity respectively. Furthermore, we define *Very Easy* benchmark as DCS with intensity 0.1 and without camera movement. *Clean* refers to the clean environment with all distractions disabled (see example images in the appendix).

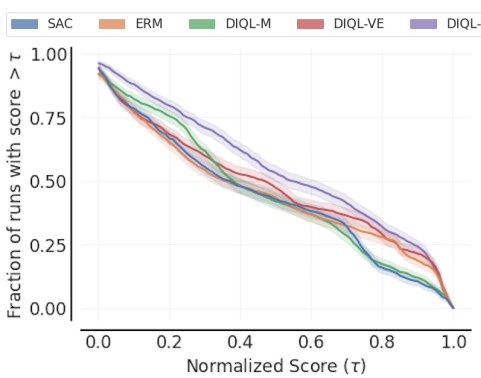

Figure 4: Performance profile aggregated over all benchmarks, tasks, 21 episodes and 4 seeds at the end of training. Shaded areas represent stratified bootstrap confidence intervals.

**Training.** We compare our methodology with several baselines. **SAC** refers to Soft Actor-Critic applied with only access with one observation at each time step. This corresponds to the usual implementation of SAC on top of a visually randomized environment. **ERM** average policy and Q-value losses of SAC over two different observations of the same step at each step, while keeping the action and reward fixed. **DIQL** refers to our methodology, which is implemented with SAC and described in section 3. Main results with DIQL, ERM and SAC are obtained by training on the *Easy* benchmark. We might refer to **DIQL-VE**, **DIQL-E** or **DIQL-M** which describe DIQL trained with the *Very Easy*, *Easy* and *Medium* benchmarks respectively to compare them. Otherwise, ERM and SAC are always trained with the *Easy* domain. For each experience, we train 4 random seeds for over 500k steps of gradient descent with Adam optimizer. We use the SAC implementation of ACME (Hoffman et al., 2020) in Jax (Bradbury et al., 2018) for faster training. Both policy and Q-network are implemented with convolutional stack followed by a MLP. The policy and Q-network only share weights of the convolution stack to compute lower-dimensional visual features.

**Evaluation.** We evaluate our models during and at the end of training over the 5 benchmarks defined above: *Clean*, *Very Easy*, *Easy*, *Medium* and *Hard*. We run evaluation episode on each benchmark every 50000 steps of training and accumulate the return over each episode. Episodic return adds up to 1000 but we normalized the score. We systematically use the *rlliable* library (Agarwal et al., 2021b) to evaluate our models, using stratified boostrap over seeds and/or tasks on the benchmark to provide robust evaluation metrics. In particular, we use the inter-quartile mean (IQM) as a robust replacement to the mean while being more sample efficient than the median. To properly test for generalization, the evaluation environment uses a different dataset of videos for the background even when training and evaluation have the same distraction difficulty.

## 4.2   RESULTS

**Aggregated metrics.** As shown in figure 3, **DIQL** largely outperforms all baselines when aggregating results over all benchmarks. One the one hand, ERM averages both policy and Q-function losses over both training domains but does not improve over plain SAC trained directly on a single distracting domain. On the other hand, DIQL-E completely outperforms these baselines for the same computational costs, thanks to the efficient use of domain randomization to enforce Q-value invariance. DIQL-E improves IQM by roughly 30 % over the baseline and is significantly above on

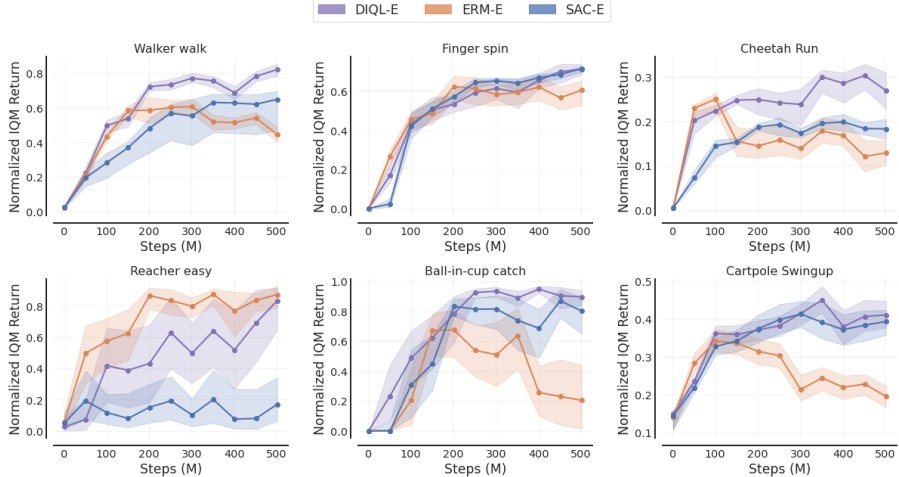

Figure 5: Sample efficiency plots for each task on the *Easy* evaluation benchmark. Models are evaluated every 50k steps and scores are aggregated over 10 episodes and 4 seeds. Shaded areas represent stratified bootstrapped confidence intervals.

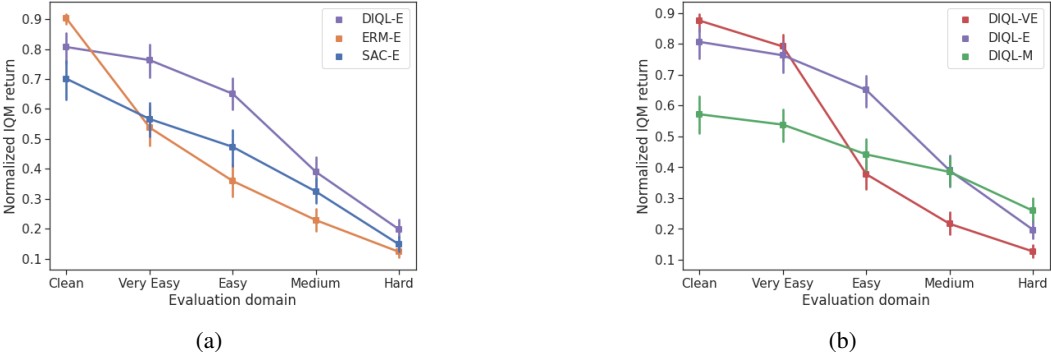

Figure 6: Evaluation at the end of training of DIQL variants and baselines across all 5 benchmark difficulties. Points are normalized IQM episodic returns, and vertical bars are bootstrapped confidence intervals. **(a):** Comparison of DIQL, ERM and SAC results trained on *Easy*. **(b):** Comparison of DIQL results trained on *Very Easy*, *Easy* or *Medium*.

the performance profile of figure 4. Figure 5 shows sample efficiency curves for each six tasks on the *Easy* benchmark (the other benchmarks are in the appendix). We observe that DIQL especially improve performance in more complex tasks like Walker walk and Cheetah run. They both have a high-dimensional action space, which hints at a potential successful transfer of our method to harder tasks in robotics. However, all tasks are overall affected by the difficulty of the benchmark and its dynamic aspect compared to a clean environment. In particular, Cartpole Swingup which should be solved easily actually shows poor performance because of the original action repeat value of 8: in between two consecutive frames seen by the agent, camera and background will have moved by 8 frames which breaks the continuity of distractions and make the task particularly hard.

**OOD generalization.** We evaluate generalization capabilities of the agent in figure 6. We seek robustness to out-of-distribution shifts and look for extrapolation in the evaluation benchmark. In this setting, DIQL is trained on each *Very Easy*, *Easy* and *Medium* benchmarks and evaluated on all five benchmarks. Because the evaluation benchmarks are ordered by distraction intensity, each evaluation domain is included in the next domain along the X axis. Evaluation domains easier or equal than the training domain are considered in-distribution: they interpolate between the *Clean* domain and the training domain. Domains strictly above the training domain are out-of-distribution: they require the model to extrapolate. Figure 6a shows final normalized IQM aggregated over tasks

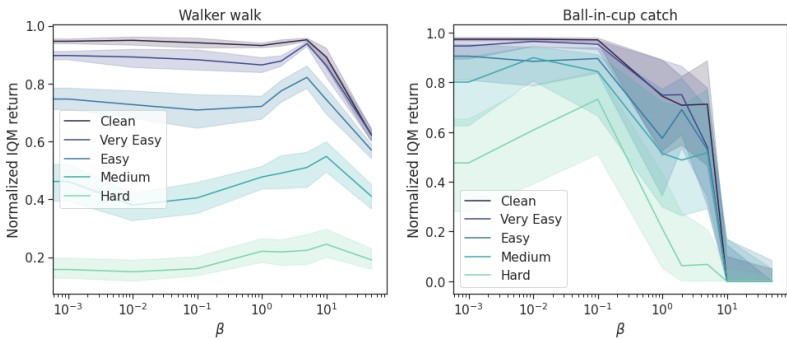

Figure 7: Evaluation score on all benchmarks for different $\beta$ values. Shaded areas are bootstrapped confidence intervals.

for each testing benchmark and all baselines and variants presented. ERM increases total return in the *Clean* benchmark. However, it fails to improve generalization to all distracting benchmarks and degrades results compared to plain SAC. Pitfalls of empirical risk minimization are to blame: it is much easier to optimize the RL objective over with non-noisy observations which leads to most capacity being dedicated to overfitting on the *Clean* domain. DIQL is able to correct this problem and redistribute the model capacity between both training environment by enforcing invariance of risks with the penalty term. Performance is lifted on all distracting benchmark as a result, but is also decreased on the *Clean* domain. Overall, the risk landscape across domains *flattens*, a phenomenon already observed and described in the seminal paper on risk extrapolation (Krueger et al., 2021).

This flattening of risk landscape is also observed by changing the training distribution used with DIQL. We trained DIQL on three different benchmark difficulties: *Very Easy*, *Easy* and *Medium*. We evaluate on all benchmarks and show results in Figure 6b. Training on harder domains helps generalization to harder benchmarks as expected, but reduce overall performance: the network capacity and training time remained constant while the training task became harder. Even though Figure 6a shows a clear improvement by DIQL in the out-of-distribution, we see in Figure 6b an expected drop of performance when going from in-distribution to OOD evaluation regime. In particular, DIQL-VE score drops by more than 50 % between *Very Easy* and *Easy* evaluations while DIQL-E score drops less than 40 % between *Easy* and *Medium*. DIQL-M only drops by rougly 30% between *Medium* and *Hard* but overall performance is lower.

The risk extrapolation term controlled by $\beta$ plays a significant role in flattening the risk space across domains as shown in Krueger et al. (2021). In particular, higher $\beta$ forces equalization of training losses of DIQL which prevents overfitting on easy domains. Figure 7 shows evaluation scores as a function of $\beta$. Each task has its own different optimal $\beta$ value depending on the task dynamics and complexity, which makes this parameter important to optimize for maximum performance.

**Ablations.** We performed a detailed ablation of $\mathcal{L}_{ERM}$ and $\mathcal{L}_{DT}$ described in section 3 and show that both term in their symmetric form are necessary for good performance across all benchmarks. Results are shown in appendix E.

## 5 RELATED WORKS

**Data augmentation.** Although not studied in this work, data augmentation is an efficient and easy to implement method to regularize models and increase generalization performance. Some methods study which type of data augmentation is better suited for reinforcement learning. Apart from the standard random shift/random crop (Yarats et al., 2020; 2021; Laskin et al., 2020b), some work explored data mixing strategies (Wang et al., 2020; Zhang & Guo, 2021; Zhou et al., 2020) inspired by mixup style data augmentation. Lee et al. (2020); Zhou et al. (2020) explored random convolutions and convolution mixing respectively. Recents works (Seo et al., 2022; Xiao et al., 2022) used patch-based masking, inspired from masked autoencoders in computer vision. Automatic adjustment of data augmentation is also studied, with Raileanu et al. (2021) using upper-confidence-bound bandit algorithm to find the transformation that maximizes return, Zhang & Guo (2021); Agarwal & Chin-

chali (2022) opting for an adversarial objective making the task as difficult as possible for the policy and Lu et al. (2020) using learned causal models to apply visual interventions. Other works study how and when data augmentation should be applied. Hansen & Wang (2021); Hansen et al. (2021) apply DA assymetrically to source and target encoder and/or Q-value functions. Ko & Ok (2021) show that data augmentation shouldn't be applied uniformly during training, while Fan et al. (2021) only use strong data augmentation for behavior cloning of an expert policy. While efficient, these methods are sub-optimal if our goal is to increase data diversity for control because data augmentation is only a post-processing transformation as explained in our paper.

**Visual domain randomization.** Contrary to data augmentation, domain randomization is directly provided by the environment, as the changes are applied at render time or even while building the scene components. Hansen & Wang (2021); Stone et al. (2021); Grigsby & Qi (2020) modify the original Deepmind Control Suite (Tassa et al., 2018) to include visual distractions, Xing et al. (2021); Zhu et al. (2020) involve robotic tasks and Ahmed et al. (2020) specifically focus on causal relationships in robotic toy environments. Domain randomization can also be integrated inside the methodology, by enforcing inductive bias to the agent such that it remains robust to visual changes. Akkaya et al. (2019) dynamically adapts the level of domain randomization to the agent's performance while Ren et al. (2020) use an adversarial objective similar to Zhang & Guo (2021). Other works run multiple versions of the same environment in parallel and use adversarial objectives Ren et al. (2020); Li et al. (2021) or enforce invariance between environments Zhao & Hospedales (2021); James et al. (2019); Zhang et al. (2020).

**Invariant representation learning** is another approach to ensure good generalization across visual perturbations and is usually implemented as an auxiliary self-supervised task to the reinforcement learning algorithm. Zhang et al. (2021); Agarwal et al. (2021a); Bertran et al. (2022) use bisimulation and behavioural similarity to learn invariant representations. Other works use the causal inference framework to isolate causal feature sets. Zhang et al. (2020) learns a model of the environment with invariant causal prediction (Peters et al., 2016) in the block-MDP setting by unrolling multiple versions of the same environment in parallel while Sonar et al. (2021) uses invariant risk minimization (Arjovsky et al., 2019) across multiple domains to learn representations that are invariant to action prediction. Lu et al. (2020) combined data augmentation with counterfactuals to learn a structured causal model with an adversarial GAN-like objective. Li et al. (2021) also uses an adversarial objective combined with gradient reversal to learn a representation that is less predictive of interventions on style variable. Mozifian et al. (2020) combined bisimulation and risk extrapolation (Krueger et al., 2021) to learn robust representations under domain randomization for robotic tasks. We differ from this line of work as DIQL is model-free and does not use an auxiliary representation learning loss.

## 6 CONCLUSION

We presented in this paper DIQL (Domain-invariant Q-learning), a model-free method that extends deep Q-learning for robust control from images under visual distractions. Our method combines efficient use of domain randomization with risk extrapolation to learn a domain-invariant Q-function. We demonstrated strong results and sample efficiency on the Distracting Control Suite benchmark. To our knowledge, DIQL shows state-of-the-art performance on *Easy* and *Medium* benchmarks (intensity 0.1, 0.2) with all three types of dynamic distractions at the same time, especially camera movement distractions. Due to the simplicity of implementation and low computational burden of the method compared to representation learning or model-based approaches, we advocate for a more generalized integration of domain randomization into reinforcement learning algorithms.

An obvious limitation of DIQL is its need for domain randomization, although one could adapt the loss function to work with well-suited data augmentation as in Hansen et al. (2021). Yet, recent development of simulations with increased speed and photo-realism call for systematic integration of domain randomization into RL pipelines for robust agents acting from images. Future work involve scaling the method to more realistic environment, with more interventions per step to maximize sample efficiency and testing results on real robot learning.

## REPRODUCIBILITY STATEMENT

We provide in Appendix B all hyperparameters associated with the method, implementation details for the method and the environment, as well as links to the github repositories we based our implementation on.

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

## A CONTINUOUS CONTROL WITH DIQL

**Soft Actor-Critic** extends Q-learning to continuous control with an entropy maximizing actor-critic algorithm. The policy loss is defined as follow:

$$J_\pi(\phi) = \mathbb{E}_{\mathbf{s}_t \sim \mathcal{D}} \left[ \mathbb{E}_{\mathbf{a}_t \sim \pi_\phi} \left[ \alpha \log \left( \pi_\phi \left( \mathbf{a}_t \mid \mathbf{s}_t \right) \right) - Q_\theta \left( \mathbf{s}_t, \mathbf{a}_t \right) \right] \right] = \mathbb{E}_{\mathbf{s}_t \sim \mathcal{D}} \left[ L_\pi(\mathbf{s}_t) \right]$$

The temporal-difference loss used to learn the action-value function is similar to equation 1 with an additional entropy regularization term and sampling $\mathbf{a}_{t+1} \sim \pi_\phi(\mathbf{s}_t)$:

$$Q_{\text{target}} = r_t + \gamma \left( Q_{\bar{\theta}}(\mathbf{s}_{t+1}, \mathbf{a}_{t+1}) - \alpha \log \pi_\phi \left( \mathbf{a}_{t+1} \mid \mathbf{s}_{+1} \right) \right)$$

where $\alpha$ is a temperature parameter controlling exploration and is either fixed or trainable.

Deep Q-learning in itself only implements a Q-function and can only output discrete actions. For continuous control, we adapt the invariant Q-learning loss to Soft Actor-Critic (Haarnoja et al., 2018) where the temporal-difference error is naturally replaced with the soft Bellman error. On top of augmenting the soft TD error with the DIQL loss, we average the policy loss of SAC over the two observations of the same state. In theory, an invariant Q-function would suffice to learn an invariant policy with SAC as the policy takes actions that maximizes Q-values. In practice, this is rarely true during training. We enforce the policy to also stay invariant by averaging the policy loss over both observations.

$$J_\pi^{DIQL}(\phi) = \frac{1}{2} \left( J_\pi(o_t^1) + J_\pi(o_t^2) \right)$$

where $J_\pi$ is defined just above.

## B IMPLEMENTATION

Each experience in the paper is run on 4 different seeds for reproducibility. We use the SAC implementation of ACME[1] (Hoffman et al., 2020) in Jax (Bradbury et al., 2018) for our DIQL implementation, and only change the critic and actor losses as described in section 3 and A. Both policy and Q-network are implemented with convolutional stack followed by a MLP. The policy and Q-network only share weights of the convolution stack to compute lower-dimensional visual features. The convolutional stack (or "encoder) is composed of 4 convolutional layers with 32 filters and $3 \times 3$ kernel sizes. Stride is 2 for the first convolutional layer then 1 for the rest. Outputs features are flattened and put through a linear layer to reach a final dimension of 50. Layer normalization and tanh activation is applied to the features before passing them to actor or critic's MLP. Encoder layers are initialized with delta orthogonal initialization, while all linear layers used Lecun uniform unitialization. All networks use ReLU activations units. Trainig is done with the Adam optimizer, and all hyperparameters used are described in table 1. Importantly, $\beta$ was optimized only for two tasks (Walker and Ball in cup) for compute availability reasons and kept at a constant value of 1 for others. Results can be improved for the four other tasks by finding a more optimal $\beta$ for each of them.

We use the github[2] implementation of the Distracting Control suite for the *Easy*, *Medium* and *Hard* benchmarks and only modify the *Very Easy* by removing camera movement.

[1] https://github.com/deepmind/acme
[2] https://github.com/geyang/gym-distracting-control

Table 1: Hyperparameters

| Hyperparameter | Value |
| --- | --- |
| Replay buffer size | 100000 |
| Initial collection steps | 25000 |
| Optimizer | Adam |
| Actor learning rate | 3e-4 |
| Critic learning rate | 3e-4 |
| Weight decay | 0 |
| Initial temperature $\alpha$ | 0.1 |
| Temperature learning rate | 3e-4 |
| Batch size | 128 |
| $\tau$ EMA | 5e-3 |
| Actor hidden layers | [512, 512] |
| Critic hidden layers | [512, 512] |
| Frame stacking | 3 |
| Action repeat | 8 if Cartpole; 2 if Finger, Walker; 4 otherwise |
| DIQL variance penalty $\beta$ | 5 if Walker; 0.1 if Ball in Cup; 1 otherwise |

## C DISTRACTING CONTROL SUITE

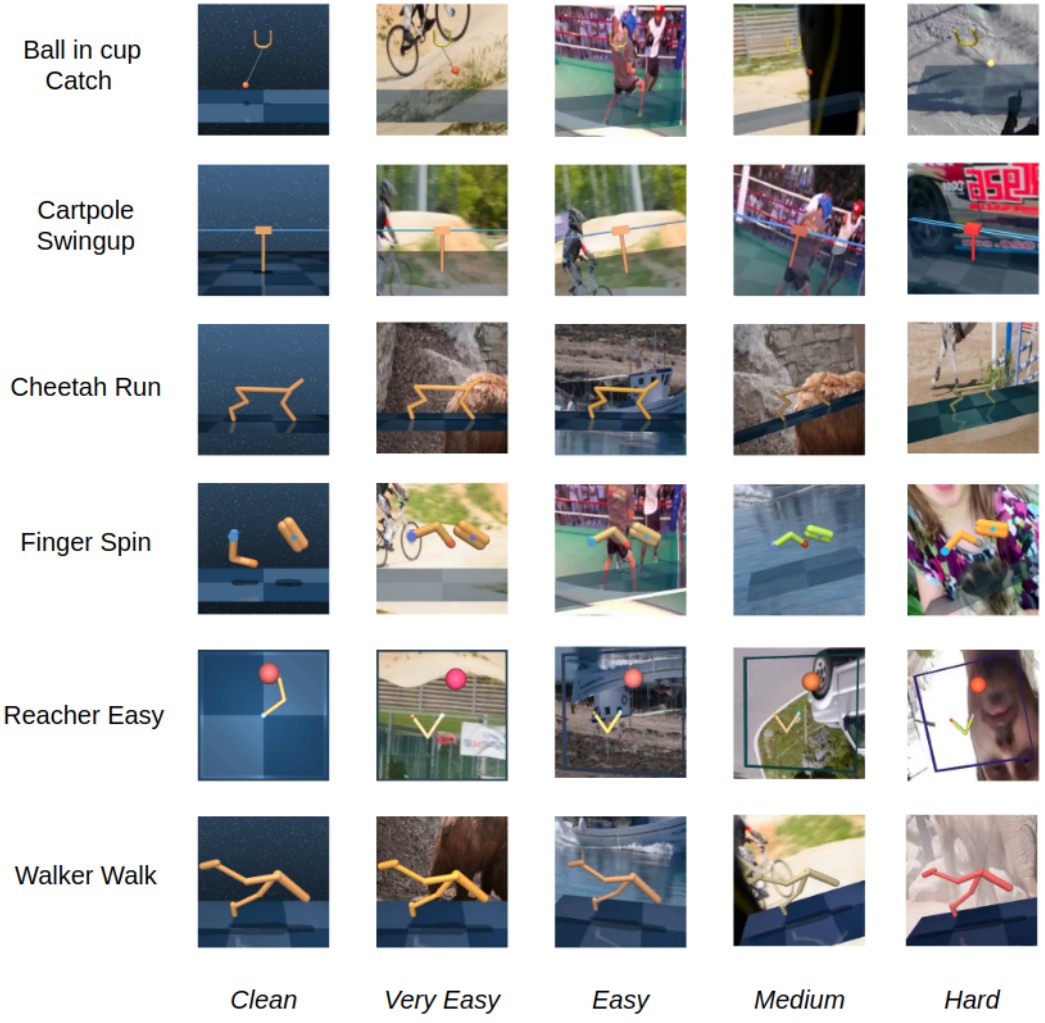

Figure 8: Distracting Control Suite tasks and evaluation benchmarks used in the paper.

# D DETAILED RESULTS

## D.1 SAMPLE EFFICIENCY CURVES

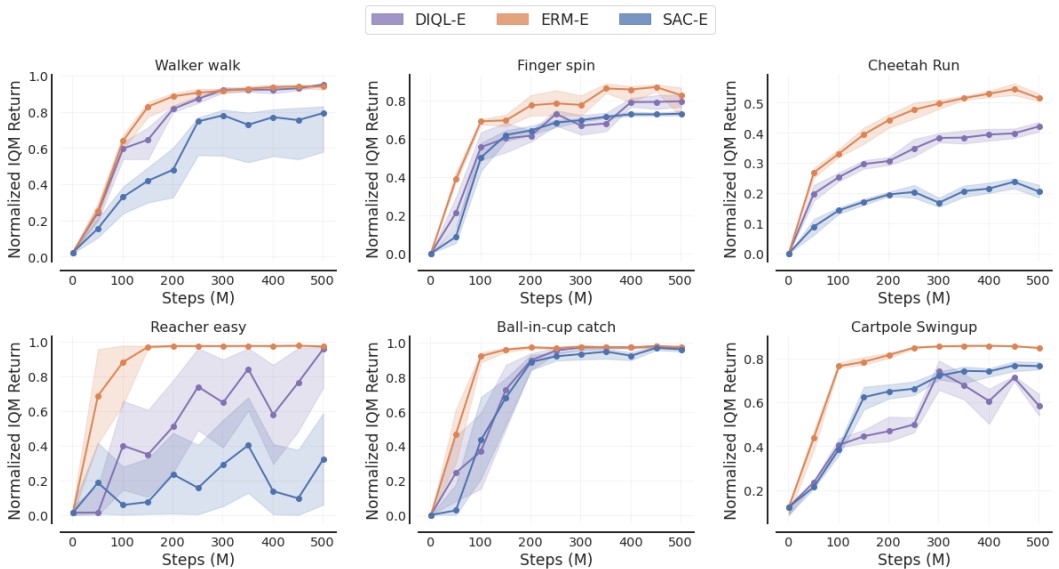

Figure 9: Sample efficiency curves on *Clean* benchmark. Line is normalized IQM and shaded area bootstrapped CI.

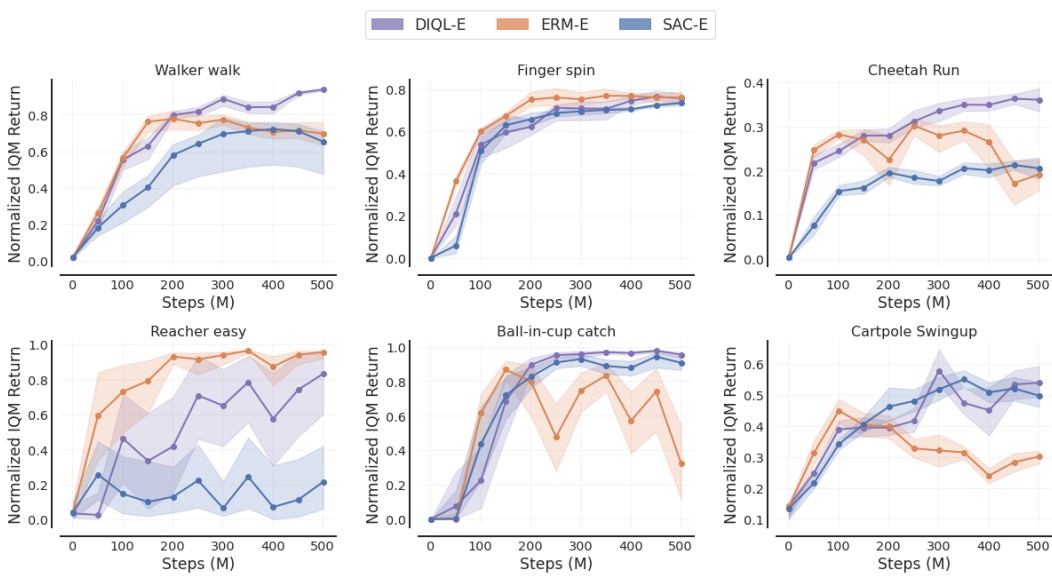

Figure 10: Sample efficiency curves on *Very Easy* benchmark. Line is normalized IQM and shaded area bootstrapped CI.

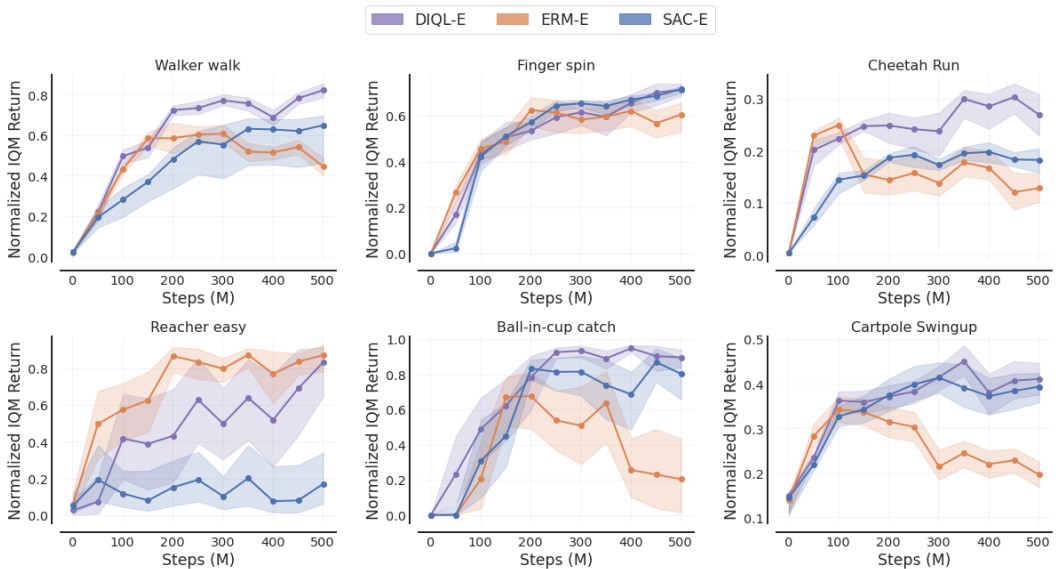

Figure 11: Sample efficiency curves on *Easy* benchmark. Line is normalized IQM and shaded area bootstrapped CI.

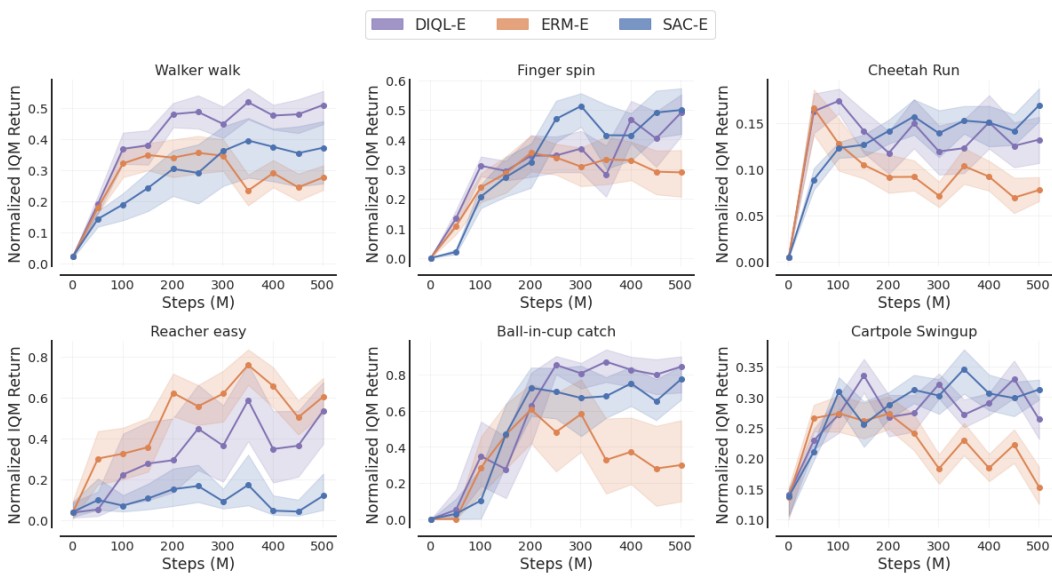

Figure 12: Sample efficiency curves on *Medium* benchmark. Line is normalized IQM and shaded area bootstrapped CI.

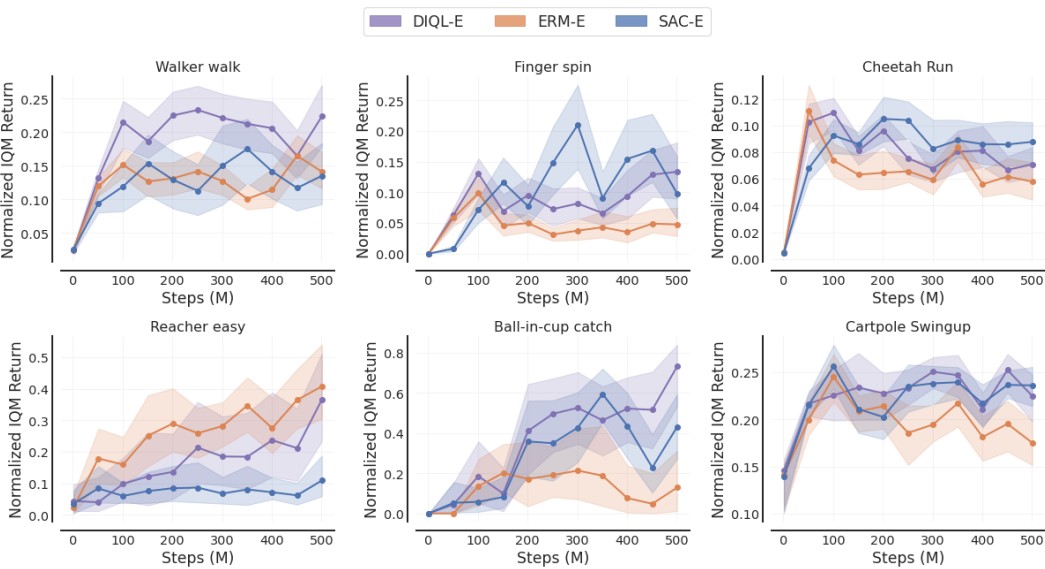

Figure 13: Sample efficiency curves on *Hard* benchmark. Line is normalized IQM and shaded area bootstrapped CI.

## D.2 GENERALIZATION

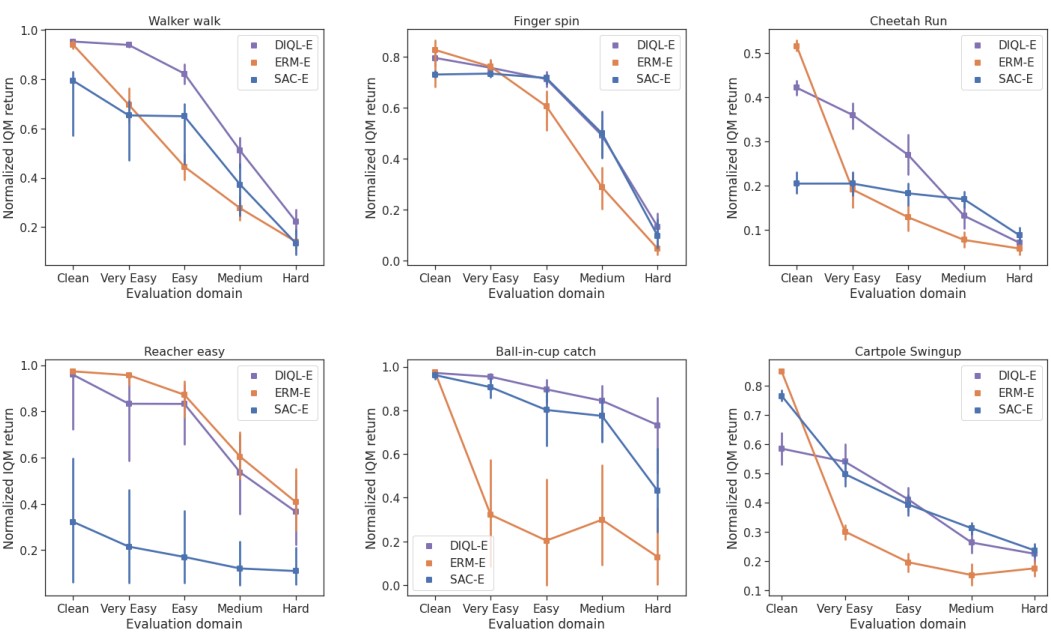

Figure 14: Evaluation at the end of training of DIQL, ERM and SAC across all 5 benchmark difficulties splitted per task. Points are normalized IQM episodic returns, and vertical bars are bootstrapped confidence intervals.

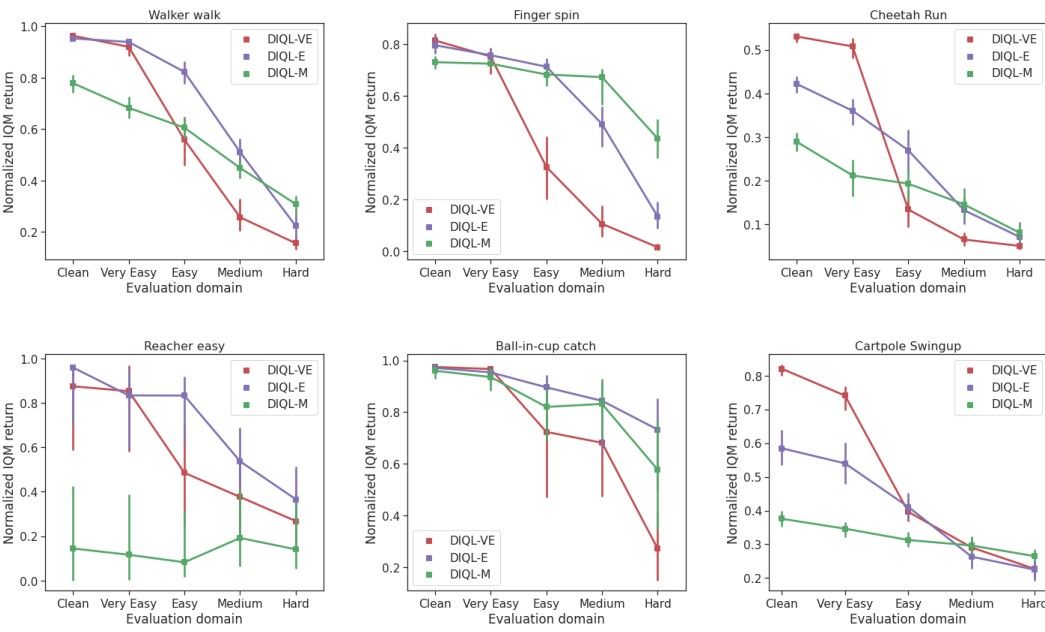

Figure 15: Evaluation at the end of training of DIQL-E, DIQL-VE and DIQL-L across all 5 benchmark difficulties splitted per task. Points are normalized IQM episodic returns, and vertical bars are bootstrapped confidence intervals.

# E ABLATIONS

We perform ablations on the ERM and DT (domain translation) terms from the DIQL loss and show that all four TD error terms are necessary for good performance. To perform the ablation, we disabled risk extrapolation by fixing $\beta$ to 0. We define the following notations for the ablations: $\mathcal{L}_{ERM}$ is composed of $ERM_1$ and $ERM_2$, $\mathcal{L}_{DT}$ of $DT_1$ and $DT_2$ (each corresponding to one of the two TD loss of each term). $ERM$ refers to $\mathcal{L}_{ERM}$. DT refers to $\mathcal{L}_{DT}$. $ERM + DT$ is equivalent to DIQL with $\beta = 0$. Training is done on the *Easy* benchmark for all ablations.

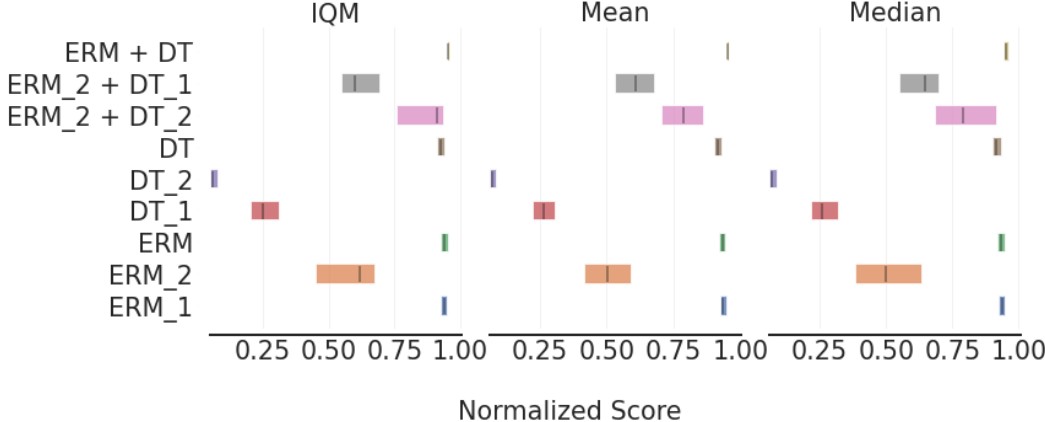

Figure 16: IQM and bootstrapped CI of ablations on *Clean*

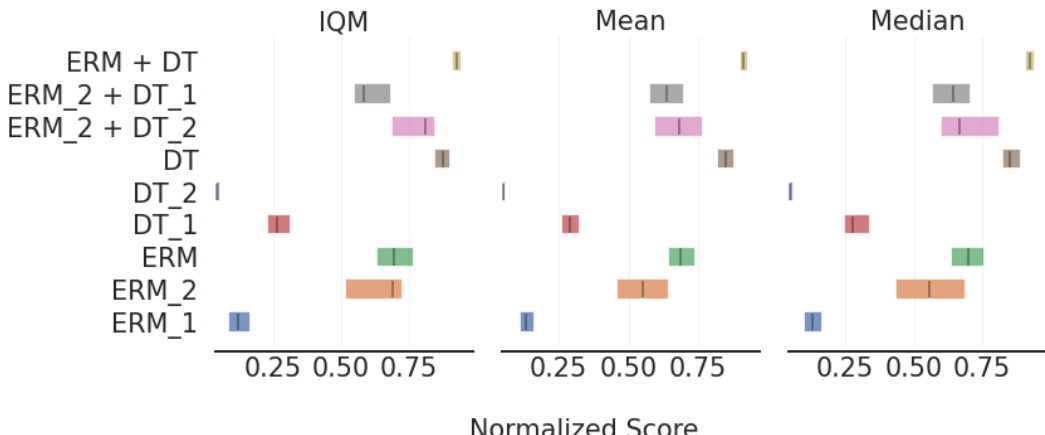

Figure 17: IQM and bootstrapped CI of ablations on *Very Easy*

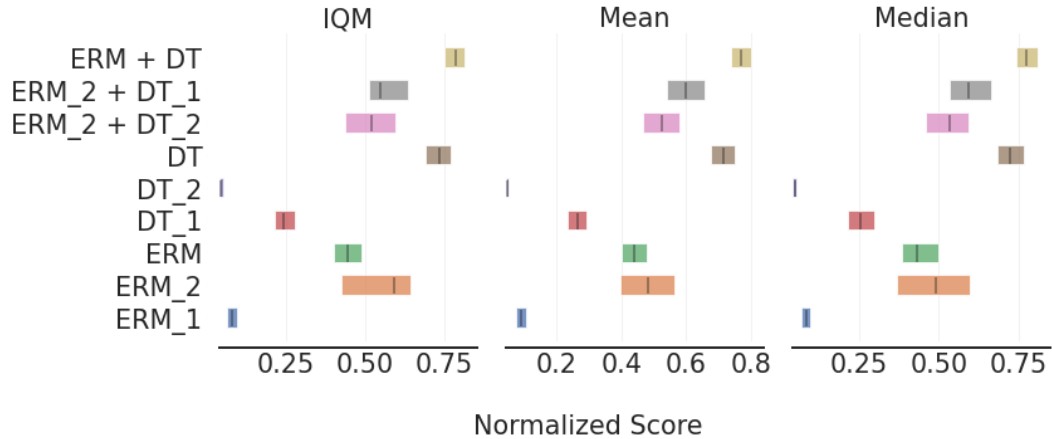

Figure 18: IQM and bootstrapped CI of ablations on *Easy*

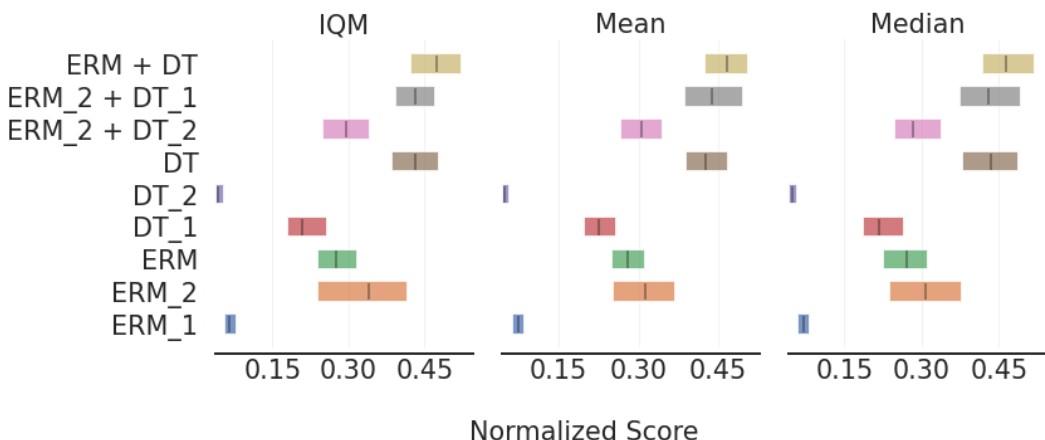

Figure 19: IQM and bootstrapped CI of ablations on *Medium*

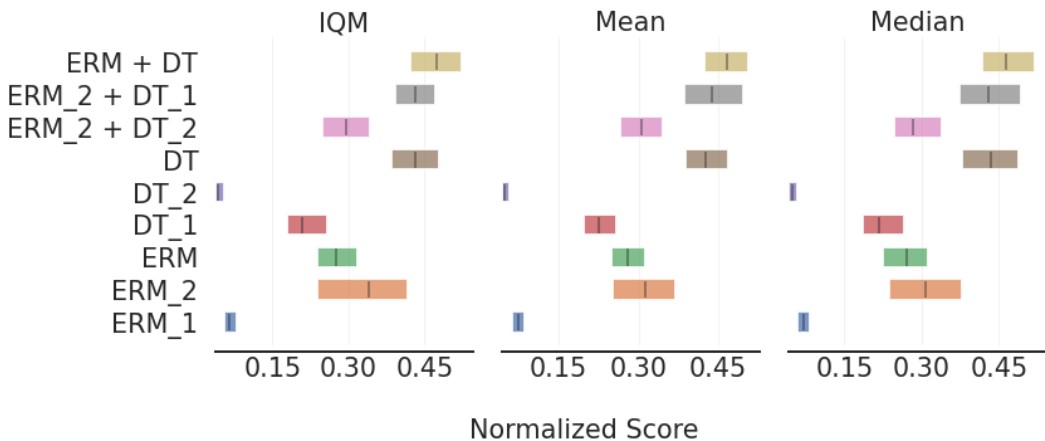

Figure 20: IQM and bootstrapped CI of ablations on *Hard*

