# OpenReview forum: "Domain Invariant Q-Learning for model-free robust continuous control under visual distractions"
_ICLR.cc/2023/Conference — Submitted to ICLR 2023_

### Official Review · Reviewer_N4B4 · 2022-10-25

**Confidence:** 4
**Correctness:** 3
**Technical Novelty And Significance:** 3
**Empirical Novelty And Significance:** 2
**Recommendation:** 3

**Clarity, Quality, Novelty And Reproducibility:**

- Clarity
    - The paper is generally well written and easy to follow. It introduces reasonable background knowledge to understand the method.
- Quality
    - The results do not seem significant enough due to reasons explained in Main Weaknesses.
- Novelty
    - There is reasonable novelty to adapt previous work on OOD generalization to domain-invariant Q-learning.
- Reproducibility
    - Experiments are run with 4 seeds. Architecture details and hyperparameters are provided in Appendix.

**Strength And Weaknesses:**

- Strengths
    - The paper is generally well written and easy to follow.
    - Ablation study shows the effectiveness of each loss term.
- Main Weaknesses
    - It seems unclear whether the problem setting is interesting or meaningful to tackle. This paper assumes access to the clean state in order to be robust to distractions. This sounds impractical, and most work in this field (e.g., [DBC](https://openreview.net/forum?id=-2FCwDKRREu), [TIA](https://proceedings.mlr.press/v139/fu21b.html), [TPC](https://proceedings.mlr.press/v139/nguyen21h.html), [Dreaming](https://arxiv.org/abs/2007.14535), [DreamerPro](https://proceedings.mlr.press/v162/deng22a.html)) does not have access to clean states when dealing with distractions. It is even unclear whether the access to clean states is necessary, as the paper does not compare to these methods.
    - The baseline is SAC with no access to the clean state. This seems to be a weak baseline, and it is not surprising that the proposed method, with access to the clean state, can outperform this baseline. A reasonable baseline could be [DrQ](https://arxiv.org/abs/2004.13649), with the augmented views replaced by the clean and noisy views.
- Questions and Minor Issues
    - Why use one clean view and one noisy view, not two noisy views? What happens when you use many noisy views?
    - It is a bit confusing to put models trained on different noise levels into one figure (e.g., Figure 3).

**Summary Of The Paper:**

The paper tackles the problem of distraction robustness and out-of-distribution (OOD) generalization in model-free RL. To this end, the paper proposes to use domain randomization combined with some modification to the Q-learning objective. Specifically, the model takes two views, one clean and one noisy. The noisy view is generated by adding random noise to the groundtruth state (as opposed to pixel-space augmentation). For each view, there are two loss terms, regressing the Q-network toward the two Q-targets computed from both views (as opposed to only a single Q-target computed from the same view). To further encourage invariance of the Q-network, the paper proposes to also minimize the variance among these loss terms, following a previous work on OOD generalization. The experiments focus on the Distracting Control Suite, with SAC being the backbone RL algorithm and main baseline. It is shown that the proposed method outperforms SAC in distracting environments, and can generalize better than SAC to larger noise during evaluation. Ablation shows that all loss terms have effect.

**Summary Of The Review:**

I recommend reject for the paper in its current form, due to potentially limited significance and weak baselines.

---

> ### Author Response · Authors · 2022-11-16
> **Author responses to Reviewer N4B4**
>
> We thank the reviewer for its comments and address them subsequently.
>
> **Q: It seems unclear whether the problem setting is interesting or meaningful to tackle.**
>
> **A:** We refer the reviewer to our global answer to all reviewers on this subject, where we explain the problem setting and why we thing it is important to consider it.
>
> **Q: This paper assumes access to the clean state in order to be robust to distractions. This sounds impractical, and most work in this field (e.g., DBC, TIA, TPC, Dreaming, DreamerPro) does not have access to clean states when dealing with distractions. It is even unclear whether the access to clean states is necessary, as the paper does not compare to these methods.**
>
> **A:** It is impractical if we place ourserlves in the real world, but is very easy to achieve during simulation. Because in the Sim2Real transfer setting, we only train in simulation and evaluate in the real world, there is nothing preventing us from using privilege information during training, as long as we don’t need it for inference. In our case, we only use the hidden state to generate multiple views where we know for certain that they correspond to the same inner state without loss of information, on the contrary to data augmentation which might destroy critical information in controllable variables (e.g relative positions and scales, orientation).
>
> The “clean state” here is just a “less noisy” observation that a truly clean state, as uncontrollable and useless visual features are still entangled  in with controllable variables. We used two different level of noise (or “clean” and “noisy” observations as called in the paper) in order to stabilize training. The benchmark being particularly hard because of simultaneous dynamic camera movement, color and video backgrounds (ie we move the camera at every step for example), using a clean state helps convergence on this benchmark in practice and makes the problem solvable. We argue however that previous to our work, no other work managed to produce non-zero performance on the medium and hard setting of the Distracting Control Suite (i.e. all 3 dynamic distractions activated together). We emphasize that such performance was reached not only during training but also during test (all figures in the paper show evaluation/test performance of the algorithm, ie with different colors, backgrounds and camera movements than during training). Importantly, the policy never had access to the clean state during evaluation but still produced strong performance in out-of-distribution settings.
>
> **Q: The baseline is SAC with no access to the clean state. This seems to be a weak baseline, and it is not surprising that the proposed method, with access to the clean state, can outperform this baseline. A reasonable baseline could be DrQ, with the augmented views replaced by the clean and noisy views.**
>
> **A:** We agree with the reviewer that SAC only is a weak baseline with respect to this problem. However, we kindly point out  that the ERM baseline described in the paper consists in practice in DrQ with K=2 and M=1 where augmentations are replaced with both noisy and clean views, making it a much stronger baseline to compare our method to, and we show that DIQL still largely outperform this baseline especially under heavy distractions settings.
>
> **Q : Why use one clean view and one noisy view, not two noisy views? What happens when you use many noisy views?**
>
> **A:** As described above, we didn’t use two noisy views because of practical considerations of this particular benchmark known to be difficult.Using a clean view and a noisy view reduces the variance of the Q-function and helps optimization. Moreover, this asymmetry of views is found a lot in self-supervised learning for computer vision with siamese networks-like methods applying strong augmentations to the source encoder and weak  However, nothing prevents us to use two noisy views in other environments and the choice here was more practical to reduce variance of this particular environment.

---

### Official Review · Reviewer_AhUs · 2022-10-25

**Confidence:** 3
**Correctness:** 3
**Technical Novelty And Significance:** 3
**Empirical Novelty And Significance:** 2
**Recommendation:** 3

**Clarity, Quality, Novelty And Reproducibility:**

The paper is clearly written, and aside from some missing experimental comparisons, the experiments are well designed and present interesting results.  The different loss terms introduced in the paper for augmenting Q-learning based on different domains seem to be novel.

**Strength And Weaknesses:**

The paper makes an interesting comparison between data augmentation and domain randomization, and suggests that RL methods can take advantage of invariance built in to different domain randomization techniques.  The method is described well and the experiments show empirical advantages of the DIQL approach.

The paper currently has the following weak points, but these may be a result of a misunderstanding on my part:
- DIQL depends on having $o^1_t$ and $o^2_t$ where each comes from a different domain, but where the underlying state $s_t$ is the same.  How can this correspondence be found?  If this is just assumed known (as it must be to generate different domains), why not give direct access to the state $s_t$ to the learning algorithm rather than the more roundabout domain-translation terms in the loss?  Many other task-relevant state representation methods I am aware of do not assume this correspondence is known, but rather must select positive and negative examples for, say, contrastive learning.
- I would have liked to see augmentation-based representation learning methods, since the paper makes a point to contrast domain randomization with data augmentation.  Additionally, some selection of bisimulation-based and contrastive methods (like the ones cited in the paper) would be good to compare against empirically.

**Summary Of The Paper:**

The paper presents an approach for augmenting Q-learning to take advantage of domain randomization, name DIQL.  By producing observations from two different domains (here defined as visual observations with different distracting perturbations, as in the distracting control suite), DIQL can train its q-network to be invariant to the sorts of perturbations that do not affect the task.

**Summary Of The Review:**

The paper presents an interesting method for learning task-relevant representations of high dimensional state.  However, I am not convinced that the method is feasible, since it requires knowledge of the ground truth state to generate pairs of observations, and this knowledge seems like it should be off-limits to any approach that attempts to learn task-relevant features.  I'm hoping I've misunderstood a main point in the paper, but if not I do not think it passes the bar for acceptance.

---

> ### Author Response · Authors · 2022-11-16
> **Author responses to Reviewer AhUs**
>
> We thank the reviewer for its comments and address each of them subsequently.
>
> **Q: DIQL depends on having ot1 and ot2 where each comes from a different domain, but where the underlying state st is the same. How can this correspondence be found? If this is just assumed known (as it must be to generate different domains), why not give direct access to the state st to the learning algorithm rather than the more roundabout domain-translation terms in the loss? Many other task-relevant state representation methods I am aware of do not assume this correspondence is known, but rather must select positive and negative examples for, say, contrastive learning.**
>
> **A:** The correspondence is indeed assume known as we are in a simulation during training. We recognize that most traditional work on representation learning for reinforcement learning do not need to generate multiple views of the same inner state, but we argue that this should not limit the assumptions made about what is available to an image-based agent during training. Speed and render quality of simulation is increasing and more work are fully training policies in simulation and directly transferring the trained policy to the real world. The development of such simulations would allow easy control over the observations during training (e.g. by running multiple copies of the environment with the same seed for dynamics but a different seed for non-controllable visual variables) and we would be able to take full advantage of the generated data by the simulation. In that sense, our work is more orthogonal than concurrent to representation learning methods as we are not focused on learning invariant representations but directly learning an invariant Q-function using the privileged information of the simulation.
>
> Regarding using directly the state, we didn’t use it as it would be equivalent to defining an auxiliary loss that would predict the inner state from the perturbed observations. This objective is not entirely aligned with the objective of minimizing a Bellman residual error and would potentially hurt convergence. The advantage of our formulation is that it is entirely composed of Bellman errors, which ensures that most gradients are aligned and that we are actually optimizing for value iteration only and not some additional self-supervised learning objective on top of it.
>
> We agree with the reviewer that comparison with other data-augmentation based methods/representation learning methods might be beneficial and plan to further investigate such comparisons in the future.
>
> **Q:  this knowledge seems like it should be off-limits to any approach that attempts to learn task-relevant features**
>
> **A:** Following our answer to the previous question, we argue that such knowledge should not be off-limits to learn task-relevant features as long as it is only used during training, for example in our case to generate multiple views with domain randomization while being sure that the inner state is the same. Data augmentation unfortunately cannot guarantee that and might destroy information in the process. Also, the exact choice of data augmentation is a difficult question to answer in reinforcement learning, whereas our method altogether circumvents the problem by letting the simulation generate different views without data augmentation. We refer to our global answer to all reviewers for a more detailed point-of-view on this topic.
>
> We hope we clarified with these answers any potential misunderstanding for the reviewer.

---

### Official Review · Reviewer_3X9n · 2022-10-30

**Confidence:** 4
**Correctness:** 3
**Technical Novelty And Significance:** 2
**Empirical Novelty And Significance:** 2
**Recommendation:** 5

**Clarity, Quality, Novelty And Reproducibility:**

Clarity issues:

1. What is $S_t, A_t, R_t, T_{t+1}, t$ when transition is defined? You want to specify that $t$ denotes a timestep.

2. what is $r(T)$? Do you mean the reward of transition $T$?

3. It says in the line below equation 3 that $L_Q$ is defined in Equation 1, however, that is not the case. You want to instead define $L_Q$ separately as:

$L_Q(s, a, r, s') = (Q_\theta(s, a) - r - \gamma \max_{a'} Q(s', a'))^2$

and then use it directly in Equation 1.

4. What is $R^i$ in Equation 2? Do you mean $R(D^i)$?

5. DQN equations are defined in terms of state but later the state is considered latent. Consider defining observations and latent state from the very beginning in Section 2.

6. How is a domain defined? Do domains differ only in how the style variable $\bar{s}$ is defined and evolves?

7. What does $X \sim \mathcal{D}$ denote in definition of $R_l(D, \theta)$? What is this distribution over the domain? Similarly, what is meant by _"training distribution is partitioned into multiple domains"_? Are we referring to distribution over transitions? While it is intuitively clear what is happening in Section 2, sometimes (as above) the exact meaning of each line is hard to understand since terms are not clearly defined.

Questions

1. For domain translation loss, why not directly consider minimizing $(Q(o^{1}_t, a-t) - Q(o^2, a_t))^2$ instead? This will ensure that the Q values agree on these pair of observations.


Related Work

Following are a few related works that address learning distractor-free representations and are not discussed:

1. _Denoised MDPs: Learning World Models Better Than the World Itself_, Wang et al., 2022
2. _Provably Filtering Exogenous Distractors using Multistep Inverse Dynamics_, Efroni et al., ICLR 2022

Minor Points:

1. _"heavy visual distractions"_: instead of saying heavy, give explanations for why they are hard

2. _"mixing..."_  -> drop "..." and start the sentence with such as or e.g.,

3. - _(Figure 1_ missing a closing bracket

4. - "(DIQL)for" lacking space


**Strength And Weaknesses:**

Strength:

1. Robust reinforcement learning (RL) is an important topic and visual distractions are one of the important things an RL agent needs to be robust to.
2. Proposed approach is simple to implement and test

Weakness:

Two main concerns are:

1. The proposed approach requires access to a pair of observations at each time step that have the same controllable state $(s)$ but different uncontrollable state $(\bar{s})$. If I understand correctly, experiments use the knowledge of how visual distractions are added to generate these observations. How will this be accomplished in practice? E.g., consider a driving scenario where the agent needs to drive. The observation space is an image that shows what is ahead of the car. On the side of the car, there can be noise from people walking or varying weather conditions. How will one generate this pair of observation that only changes these distractions but not the controllable state? This may require domain knowledge almost equivalent to knowing the state. In contrast, the domain knowledge required in computer vision pipeline, as abstracted in a data augmentation setting, is very basic and includes simple operations such as rotation and flipping.

It is also assumed that different domains differ only in the uncontrollable state noise but otherwise have the same state. This is a particular kind of domain generalization. For example, if one considers navigation in two different buildings, then even the underlying state is different.

2. The writing in this paper is very low quality. This includes mathematical notations used without definition, using words without definition, etc. See clarity section below for examples. Even though the main ideas are easy to understand for an RL audience, the paper needs serious editing for readability.

**Summary Of The Paper:**

This paper studies reinforcement learning across domains in the presence of visual distractions. A visual distraction is a part of the underlying state that is uncontrollable, i.e., its change is independent of agent's action, and does not affect the Q values. The proposed approach uses domain randomization in a Deep Q-learning setup. At each time step $t$ in an episode, one assumes access to two observations $o^1_t$ and $o^2_{t+1}$ which denote the same controllable state but different uncontrollable state. Here the superscripts 1 and 2 are used to denote the two different domains. The proposed approach includes two additional loss:

- a domain transfer loss: $L(o^1_t, a_t, r_t, o^2_{t+1}) + L(o^2_t, a_t, r_t, o^1_{t+1})$, where $L$ is the temporal difference loss defined by $L(o, a, r, o') = (Q(o, a) - r - \gamma \max_{a'} Q_{target}(o', a'))$. This is done to ensure that Q functions remain invariant to the uncontrollable noise. This claim is not proven.

- a variance loss that computes variance of $\{L(o^1_t, a_t, r_t, o^2_{t+1}), L(o^2_t, a_t, r_t, o^1_{t+1})\}$. This is used so that losses for both domains are roughly of the same scale. It is not explained why this would happen but a reference to Krueger et al., 2021 is made.

Experiments on Distracting Control Suite (Stone et al, 2021) are presented and show promise of the approach.


**Summary Of The Review:**

I currently lean towards weak reject based on two concerns: (i) strong assumptions on generating a pair of observations with the same state but different style variable, and (ii) issues with writing. For (i), I would like to understand how to satisfy this assumption in practice outside simulations. I'll change my score based on author response (and paper revision) to these concerns.

---

> ### Author Response · Authors · 2022-11-16
> **Author responses to Reviewer 3X9n**
>
> We thank the reviewer for it’s extensive review addressing the quality of writing and raising important questions about the scope of the paper.
>
> We address the detailed comments about the clarity and quality of writing by submitting an edited version of the paper that corrects the issues pointed out by the reviewer.
>
> We then address the following main points brought by the reviewer:
>
> **Q: a domain transfer loss [...] This is done to ensure that Q functions remain invariant to the uncontrollable noise.This claim is not proven.**
>
> **A:** We draw inspiration from work in Generative Adversarial Netowrks, in particular CycleGAN which defines a cycle-consistency loss for style transfer between two domain. Our loss is similar in idea, as enforce the Bellman operator to be resilient to domain change by using the target of the other domain. Cycle consistency is enforced by symmetrizing the loss.
>
> **Q: a variance loss [...] This is used so that losses for both domains are roughly of the same scale. It is not explained why this would happen but a reference to Krueger et al., 2021 is made.**
>
> **A :** We introduce a variance minimization loss to prevent one objective from dominating the others. This prevents most capacity of the model being dedicated to only one domain, which would hurt generalization. In practice, when using a clean and noisy view, the model has the tendency to overfit to the clean domain and this loss redistributes equally the capacity to all training domains reducing as a result the out-of-distribution risk as defined in Section 2.
> We also show in section 3 that minimizing such loss allows the model to perform causal discovery of the hidden controllable variables, using Theorem 1 from Krueger et. al.
>
> **Q: How will this be accomplished in practice?**
>
> **A:** Our work places itself in the zero-shot Sim2Real transfer framework, where we want to train policies in simulation and test in reality without any additional fine-tuning. This context is now becoming common in reinforcement learning research in robotics. This method cannot be used for training in the real world but is designed to be used for training in simulation, before deploying the policy in the real world. We argue that designing methods specifically to be trained in simulation is not a problem definition issue and is actually crucial regarding the progress of simulation for reinforcement learning. Multiple works (eg. [1], [2] among others) already leverage rendering capacities and differentiable simulation for reinforcement learning to fully exploit available data. We defend that using privileged information from the simulation is not off-limits anymore for reinforcement learning considering the rapid development of simulations.
> Regarding the specific use case of autonomous driving, we believe training partly in simulation is not out-of-the question when looking at progress in simulation for this use case (eg. CARLA benchmark, or NVIDIA Drive Sim more recently with features such as transforming raw data captured by real captors on a car into a photorealistic digital twin of the scene and directly allowing scene manipulation in simulation similar to domain randomization).
>
> **Q: In contrast, the domain knowledge required in computer vision pipeline, as abstracted in a data augmentation setting, is very basic and includes simple operations such as rotation and flipping.**
>
> **A:** We agree with the reviewer that computer vision pipelines do not assume domain knowledge and use basic data augmentations to regularize RL, but we point out our work is not concurrent but rather orthogonal to traditional representation learning techniques based solely on data augmentation. The point of DIQL is to relax the assumptions made about the data the algorithm has access to and let the simulation do the heavy work of producing diverse and adversarial scenes for the training algorithm. We postulate that such assumption is again reasonable given progress in simulation quality and speed (Isaac Sim, Brax...).

---

> > ### Author Response · Authors · 2022-11-16
> > **Author responses to Reviewer 3X9n**
> >
> > **Q: For domain translation loss, why not directly consider minimizing (Q(ot1,a−t)−Q(o2,at))2 instead? This will ensure that the Q values agree on these pair of observations.**
> >
> > **A :** Minimizing such loss would correspond to having an « auxiliary » task whose objective is different from minimizing a Bellman residual error in deep Q-learning. While work using representation learning accelerates training by designing auxiliary task whose gradient directions are somewhat to the ones of the RL objective, it can sometimes be difficult to optimize and as shown by [4] can actually hurt performance in some case due to gradient conflicts between the RL and auxiliary objectives. Value-base RL methods rely entirely on the quality of the Q-function approximation. If the Q-function outputs are accurate (and invariant in our case), then the policy will naturally be good/invariant. Our work tackles the root cause of the lack of robustness to visual distractions by making the Q-value invariant to such distractions. More formally, our loss directly promotes an invariant predictor (the Q-function) as opposed to an invariant representation which might not be sufficient enough for sample efficient RL [3, 4]. [5] also show that invariant prediction should be preferred over invariant representation learning from a theoretical point of view.
> >
> > [1] J. Lv, Y. Feng, C. Zhang, S. Zhao, L. Shao, and C. Lu, *SAM-RL: Sensing-Aware Model-Based Reinforcement Learning via Differentiable Physics-Based Simulation and Rendering*.  2022
> >
> > [2] P. Ma, T. Du, J. B. Tenenbaum, W. Matusik, and C. Gan, *RISP: Rendering-Invariant State Predictor with Differentiable Simulation and Rendering for Cross-Domain Parameter Estimation.* ICLR 2022
> >
> > [3] S. S. Du, S. M. Kakade, R. Wang, and L. F. Yang, *Is a Good Representation Sufficient for Sample Efficient Reinforcement Learning? * ICLR 2020
> >
> > [4] X. Li, J. Shang, S. Das, and M. S. Ryoo, *Does Self-supervised Learning Really Improve Reinforcement Learning from Pixels?* 2022
> >
> > [5] M. Koyama and S. Yamaguchi, *When is invariance useful in an Out-of-Distribution Generalization problem ?*, 2021

---

### Author Response · Authors · 2022-11-16
**Addressing the scope and general interest of the paper**

We thank the reviewers for their contributions and address jointly here the the legitimate questions raised about the assumptions made in the paper regarding the required knowledge of the environment state to use the method in practice.

We want to clarify the point that we are trying to make in the paper, as we feel like we haven’t been clear enough about why the presented method is interesting, which led to justifiable confusion from multiple of the reviewers. The overall objective of such method is to improve Sim2Real transfer of RL policies trained in **simulation**, with robotics applications for example. In practice, we formulate the Sim2Real transfer problem as a visual domain transfer problem, allowing us to easily evaluated the method on a simple benchmark like Distracting Control Suite. **We argue that training policies in realistic simulation with domain randomization is a non-negligible and growing trend in the reinforcement learning and robotics community, as massive progress in simulation speed, accuracy and visual quality are observed recently.** Zero-shot transfer of visual policies trained fully in simulation is now possible.

The main problem that is tackled in this paper is the difficulty of training policies in simulation with intense visual perturbations (with the objective of robustness to perturbations) through (visual) domain randomization. Naïve use of extensive domain randomization might increase the variance of the problem preventing convergence during training and generalization at test time when performing Sim2Real transfer. This fact was illustrated with the weak baseline of SAC.

We then proceed by showing that **simple but careful integration of domain randomization inside algorithms to promote invariance is very effective** compared to the simplicity of the approach without requiring additional representation learning. This was illustrated by the comparison with the baseline **ERM**. ERM averages TD losses over both clean and noisy domains, making it a fair baseline to compare as it is both an ablation of our method and is equivalent to DrQ with augmentations replaced by clean and noisy views.

We argue that the positive results shown in the paper and pointed out by the reviewers should motivate research into more native and intrinsic integration of visual domain randomization into new RL algorithms for more sample-efficient and robust reinforcement learning, leveraging progress in simulation developments.

---

### Decision · Program_Chairs · 2023-01-20

**Decision:**

Reject

**Justification For Why Not Higher Score:**

All reviewers agree that the assumption is too strong or unrealistic.

**Justification For Why Not Lower Score:**

N/A

**Metareview: Summary, Strengths And Weaknesses:**

The paper proposes a Q-learning algorithm that combines domain randomization to achieve distraction robustness and out-of-distribution (OOD) generalization. To encourage invariance, an auxiliary loss for variance minimization is introduced from a previous work. The experiments are performed on Distracting Control Suite and SAC is chosen as the baseline. It is shown that the proposed method performs better than SAC under visual distraction.

Strength. The paper is easy to follow. The ablation study is reasonable.

Weakness. The main weakness is the assumption that the ground truth state is available. All three reviewers raised a concern about this. The authors answered that it's possible because they assume that the access to a simulator where the ground truth state is accessible. But, the compared baseline SAC does not use this clean state. So, the comparison is unfair. In addition, it is clearly an evidence that the paper is poorly written, making all three reviewers confused.